spintronics

super-Poisson noise, suppression, spin–orbit coupling, quantum dot

**Author for correspondence:**
Zhimei Wang
e-mail: 120705547@qq.com

# Suppression of shot noise in a spin–orbit coupled quantum dot

Zhimei Wang, Lijun Mao, Naitao Xue and Wenting Lu

Department of Physics, Taiyuan Normal University, Jinzhong, Shanxi 030619, People's Republic of China

 ZW, 0000-0001-6431-4939

We study theoretically the transport properties of electrons in a quantum dot system with spin–orbit coupling. By using the quantum master equation approach, the shot noise and skewness of the transport electrons are calculated. We obtain super-Poisson noise behaviour by investigating the full counting statistics of the transport system. We discover super-Poisson behaviour is more obvious with the spin polarization increasing. More importantly, we discover the suppression of shot noise induced by spin–orbit coupling. The value of shot noise is gradually decreasing when spin–orbit coupling strength increases.

## 1. Introduction

The quantum transport in low-dimensional systems has attracted a great deal of attention, especially spin-polarized transport, also known as spintronics [1,2]. Employing spin is the major goal of spintronics [3] for designing electrical and optoelectronic devices. There are some profound theoretical and experimental achievements in this field in the recent years [4]. The spin–orbit coupling in solids has become a major theme in condensed matter physics [5] due to its contribution to fascinating phenomena, such as the spin Hall effect in non-magnetic materials [6], an unconventional order-parameter symmetry [7,8].

A considerable amount of theoretical and experimental work has also been performed regarding the spin–orbit coupling effect in mesoscopic systems [9–12].

Electron transport through nanostructures is a stochastic process. The transport process and the statistics of transferred charges can be strongly influenced by electron–electron interactions and quantum correlations. Full counting statistics [13,14] involves the distribution of the number of transferred charges. Full counting statistics is an effective and powerful technique in quantum statistical mechanics [15,16]. Conventional transport measurements mainly focused on the average current (the first order cumulant). In mesoscopic physics, the current

voltage characteristics alone are generally insufficient for determining all transport properties. Full counting statistics of electronic transport through nanoscopic systems can provide more information on the correlations in molecular quantum dot (QD) systems [17,18]. To date, there has been more theoretical work on this concept and fewer experimental applications, which provides extra motivation to explore full counting statistics of a molecular QD system with an external spin. Additionally, the second-order cumulant (shot noise) and higher-order cumulants can reveal more novel phenomena, such as quantum coherence, disorder, entanglement and dissipation [19]. It turns out that counting statistics is extremely important in experimental physics as this approach has the capability to measure high-order cumulants and the entire distribution of transferred charges.

It is of great interest to observe super-Poisson noise as this measure can identify additional properties of unusual transport mechanisms [20–24]. The super-Poisson-noise behaviour can be reflected in dynamical channel blockade systems [25,26], dynamical spin blockade systems [27,28], superconductor-normal-superconductor systems [29–32], bistability [33], cotunneling [34–36] and electron–phonon interaction of the shuttle systems [37].

In this paper, we mainly investigate the suppression of shot noise through the transport system. At present, there are relatively few studies on this. Therefore, it is of great scientific significance to explore the effect of spin–orbit coupling on the transport system, especially for the design of spin devices.

## 2. Model description

The Hamiltonian of the transport system with spin–orbit coupling is governed by

$$\hat{H} = \hat{H}_D + \hat{H}_{\text{Leads}} + \hat{H}_T,$$ (2.1)

$$\hat{H}_D = \sum_{\sigma j} \varepsilon_j d_{j\sigma}^\dagger d_{j\sigma} + \sum_\sigma i\sigma\alpha_{\text{so}}(d_{2\sigma}^\dagger d_{1\sigma} - \text{H.c})$$
$$+ \sum_{\sigma\sigma'} U n_{1\sigma} n_{2\sigma'},$$ (2.2)

$$\hat{H}_{\text{Leads}} = \sum_{k\sigma} (\varepsilon_{Lk} c_{Lk\sigma}^\dagger c_{Lk\sigma} + \varepsilon_{Rk} c_{Rk\sigma}^\dagger c_{Rk\sigma}),$$ (2.3)

$$\hat{H}_T = \sum_{k\sigma} [\Omega_{Lj} c_{Lk\sigma}^\dagger d_{j\sigma} + \Omega_{Rj} c_{Rk\sigma}^\dagger d_{j\sigma} + \text{H.c.}],$$ (2.4)

where $\hat{H}_D$ and $\hat{H}_{\text{Leads}}$ are the Hamiltonians in the QD and the leads, respectively. The spin–orbit coupling Hamiltonian is $\hat{H}_{\text{so}} = \sum_\sigma i\sigma\alpha_{\text{so}}(d_{2\sigma}^\dagger d_{1\sigma} - \text{H.c})$ ($\alpha_{\text{so}}$ is the spin–orbit coupling constant and spin-index $\sigma = \uparrow, \downarrow$) [38]. $\hat{H}_T$ shows the electron tunnelling between the leads and the levels, of which $\Omega_{L,R}$ is relatively weak. Thus we can regard it as perturbation.

In the above Hamiltonians, $c_{Lk\sigma,Rk\sigma}^\dagger(c_{Lk\sigma,Rk\sigma})$ with momentum $k$ and lead-$L$, $R$ denotes the electron creation (annihilation) operators corresponding to the lead reservoirs and $d_{j\sigma}^\dagger(d_{j\sigma})$ with $j = 1$, 2 describes the electron creation (annihilation) operators at the QD-$j$. $\varepsilon_{Lk}$ and $\varepsilon_j$ indicate the corresponding energy levels. $n_{j\sigma} = d_{j\sigma}^\dagger d_{j\sigma}$ indicates the occupation number operator. $U$ describes Coulomb interaction among electrons, where $U$ is very strong. The last term denotes the electron interaction at the same level. $\Omega_{Lj\sigma,Rj\sigma}$ or $\Gamma_{Lj\sigma,Rj\sigma} = 2\pi g_{L\sigma,R\sigma}|\Omega_{Lj,Rj}|^2$, for latter use, indicate the coupling strength between the leads and levels. Here, $g_{L,R}$ represent the density of states of the leads. The spin polarization is indicated as $p = (g_\uparrow - g_\downarrow)/(g_\uparrow + g_\downarrow)$. Thus $\Gamma_{L\uparrow(\downarrow)} = \Gamma_L(1 \pm p)$ and $\Gamma_{R\uparrow(\downarrow)} = \Gamma_R(1 \pm p)$. In the current study, where there is no explicit specification, we focus on the strong Coulomb blockade regime, where available occupation states are allowed, i.e. $|0, 0\rangle$, $|\uparrow, 0\rangle$, $|\downarrow, 0\rangle$, $|0, \uparrow\rangle$ and $|0, \downarrow\rangle$. Thus the eigenvalues and eigenstates of system are obtained as follows: $E_{00} = 0$, $\psi_{00} = |0, 0\rangle$, $E_\sigma^\pm = (\varepsilon_1 + \varepsilon_2 \mp \sqrt{\Delta^2 + 4\alpha_{\text{so}}^2})/2$, $|\psi_\sigma^\pm\rangle = \varepsilon_{1\sigma}^\pm|\sigma, 0\rangle + \varepsilon_{2\sigma}^\pm|0, \sigma\rangle$, $E_{\sigma\sigma'} = \varepsilon_1 + \varepsilon_2 + U$, $\psi_{\sigma\sigma'} = |\sigma, \sigma'\rangle$ ($\sigma = \uparrow, \downarrow, \sigma' = \uparrow, \downarrow$), the above eigen states of the system are normalized, and normalized coefficients with $\Delta = \varepsilon_1 - \varepsilon_2$ are given as follows (the detailed calculation process about eigenvalues and eigenstates shown in the electronic supplementary material):

$$\varepsilon_{1\sigma}^\pm = \frac{(\mp i)*\left(\Delta \mp \sqrt{\Delta^2 + 4\alpha_{\text{so}}^2}\right)}{\sqrt{\left(\Delta \mp \sqrt{\Delta^2 + 4\alpha_{\text{so}}^2}\right)^2 + 4\alpha_{\text{so}}^2}}$$

and

$$\varepsilon_{2\sigma}^{\pm} = \frac{2\alpha_{so}}{\sqrt{\left(\Delta \mp \sqrt{\Delta^2 + 4\alpha_{so}^2}\right)^2 + 4\alpha_{so}^2}}.$$

# 3. Full counting statistics of the transport system

In the above Hamiltonian, we re-express $\hat{H}_T = \hat{H}'$. In the following calculation process, we regard $\hat{H}'$ as perturbation. The evolution of the system's reduced density matrix equation is obtained in the case of the second-order Born approximation [39]

$$\frac{\partial \rho(t)}{\partial t} = -iL\rho(t) - \int_0^t dt' \left\langle L'(t)\widetilde{G}(t, t') \right.$$
$$\left. \times L'(t')\widetilde{G}^\dagger(t, t')\right\rangle \rho(t), \tag{3.1}$$

where $L(t)$ means the Liouvillian superoperators and $L(t)\rho(t) = [H(t), \rho(t)]$, $L'(t)\rho(t) = [H'(t), \rho(t)]$. $\widetilde{G}(t, t')$ represents the Green function of the Liouvillian space. $\partial\widetilde{G}(t, t')/\partial t = -iL(t)\widetilde{G}(t, t')$, $\widetilde{G}(t, t')\rho(t) = G(t, t')\rho(t)G^\dagger(t, t')$. $\langle L'(t)\widetilde{G}(t, t')L'(t')\widetilde{G}^\dagger(t, t')\rangle = Tr_B[L'(t)\widetilde{G}(t, t')L'(t')\widetilde{G}^\dagger(t, t')\rho_B]$ with $\rho_B$ the density matrix of the electron reservoirs.

Here, equation (3.1) is for all degrees of freedom of the electrode. For the purpose of describing transport characteristics more effectively, the number of electrons emitted from the left lead and arriving at the right lead needs to be tracked. Therefore, the entire Hilbert space will be classified as follows: $B^{(0)}$ (no electron arrived at the collecting electrode), $B^{(n)}$ (the electron numbers $n$ arrived at the collecting electrode). Thus, the entire Hilbert space is described as $B = \oplus_n B^{(n)}$.

Based on the above classification, the average over electrode in whole Hilbert space for equation (3.1) will be substituted for the average over electrode in the subspace. A conditional master equation is formulated as follows:

$$\frac{\partial \rho^{(n)}(t)}{\partial t} = -iL(t)\rho^{(n)}(t) - \int_0^t dt' Tr_{B^{(n)}}[L'(t)$$
$$\times \widetilde{G}(t, t')L'(t')\widetilde{G}^\dagger(t, t')\rho_T(t)], \tag{3.2}$$

where $\rho^{(n)}(t) = Tr_{B^{(n)}}[\rho_T(t)]$ is the conditional reduced density matrix. Before further derivation, two physical considerations are implemented as follows: (i) The conventional Born approximation $\rho_T(t) = \rho(t)\rho_B$ is replaced with $\rho_T(t) = \sum_n \rho^{(n)}(t) \otimes \rho_B^{(n)}$. Here, $\rho_B^{(n)}$ is the density operator of electrode reservoirs, which is connected with $n$ (electron numbers arrived at the right lead). In addition, the orthogonality between reservoir states of different subspaces determines the reserved item from $\rho_T(t)$. (ii) Because of closure property of the transport circuit, the electrons which tunnelling to the right electrode will return to the left electrode via the external circuit. On the other side, the rapid relaxation processes of the reservoirs will result in a quick recovery to local thermal equilibrium determined by the chemical potentials. For this reason, the density matrix of electron reservoirs $\rho_B^{(n)}$, $\rho_B^{(n\pm1)}$ are replaced by $\rho_B^{(0)}$.

With the above two assumptions, the quantum master equation for particle numbers is written as follows:

$$\frac{\partial \rho^{(n)}(t)}{\partial t} = -iL\rho^{(n)}(t) - \frac{1}{2}\{\sum_{\sigma j\alpha}[d_{j\sigma}^\dagger A_{j\alpha\sigma}^{(-)}\rho^{(n)}(t)$$
$$+ \rho^{(n)}(t)A_{j\alpha\sigma}^{(+)}d_{j\sigma}^\dagger + H.c.] - \sum_{\sigma j}[A_{jL\sigma}^{(-)}\rho^{(n)}(t)d_{j\sigma}^\dagger$$
$$+ A_{jR\sigma}^{(-)}\rho^{(n-1)}(t)d_{j\sigma}^\dagger + d_{j\sigma}^\dagger\rho^{(n)}(t)A_{jL\sigma}^{(+)}$$
$$+ d_{j\sigma}^\dagger\rho^{(n+1)}(t)A_{jR\sigma}^{(+)} + H.c.]\}, \tag{3.3}$$

spectral functions $A_{j\alpha\sigma}^{(+)} = \sum_{i\sigma'}\Gamma_\alpha^{ji}n_\alpha^{(+)}d_{j\sigma'}$, $A_{j\alpha\sigma}^{(-)} = \sum_{i\sigma'}\Gamma_\alpha^{ji}n_\alpha^{(-)}d_{j\sigma'}$. Here, $\Gamma_\alpha^{ji} = 2\pi g_\alpha \Omega_{\alpha j}\Omega_{\alpha i}$, $n_\alpha^{(+)} = f_\alpha$, $n_\alpha^{(-)} = 1 - f_\alpha$. $f_\alpha$ denotes electron Fermi function of lead-$\alpha$ ($\alpha = L, R$) with $f_\alpha = 1/(1 + e^{(\epsilon - \mu_\alpha)/KT})$ with Boltzmann constant $K$ and temperature $T$. $\mu_\alpha$ describes chemical potential with $\mu_L = -\mu_R = V/2$.

In this paper, the zero-frequency shot noise and skewness of the system will be studied according to the particle-solved master equation. In the quantum transport, the full counting statistics can completely describe the transport properties which provide all information of probability distribution $p(n, t)$ that there are $n$ electrons tunnelling to the right lead. However, in the actual calculation process, instead of directly deriving $p(n, t)$ by the quantum master equation, we use the cumulant generating function(CGF). Mathematically, the CGF $F(\chi)$ is defined as follows:

$$F(\chi) = -\ln \sum_n p(n, t)e^{in\chi}, \tag{3.4}$$

where $\chi$ is the so-called counting field, and $p(n, t) = Tr\rho^{(n)}(t)$. The $k$th cumulant is written

$$C_k = -\left(-i\frac{\partial}{\partial\chi}\right)^k F(\chi)\Big|_{\chi \to 0}. \tag{3.5}$$

The first three cumulants correspond to the transport properties of the system when time is long enough. The first cumulant shows the average current: $\langle I \rangle = C_1/t$, the second cumulant describes the zero-frequency shot noise: $S(0) = 2e^2(\overline{n^2} - \overline{n}^2)/t = 2e^2 C_2/t$ and the third cumulant indicates the skewness: $C_3 = \overline{(n - \overline{n})^3}$, here $\overline{(\ldots)} = \sum_n (\ldots)p(n, t)$. Moreover, the shot noise and the skewness are described by Fano factor $F_a = C_2/C_1$ and $S_k = C_3/C_1$, with a super-Poisson indicated by $F_a > 1$, and a sub-Poisson indicated by $F_a < 1$. Now, we define the following formulation:

$$S(\chi, t) = \sum_n \rho^{(n)}(t)e^{in\chi}, \tag{3.6}$$

furthermore, it is obvious that

$$e^{-F(\chi)} = Tr[S(\chi, t)]. \tag{3.7}$$

The quantum master equation can be re-expressed

$$\dot{\rho}^{(n)} = A\rho^{(n)} + C\rho^{(n+1)} + D\rho^{(n-1)}, \tag{3.8}$$

so $S(\chi, t)$ satisfies

$$\dot{S} = AS + e^{-i\chi}CS + e^{i\chi}DS = L_\chi S, \tag{3.9}$$

here $S$ is column matrix, $A$, $C$ and $D$ are square matrices. Then

$$F(\chi) = -\lambda(\chi)t, \tag{3.10}$$

where $\lambda(\chi)$ is the eigenvalue of the Liovillian operator $L_\chi$. The eigenvalues of the matrix $L_\chi$ have the expansion $\lambda = \lambda_0 + \lambda_1\chi + \lambda_2\chi^2 + O(\chi^3)$. We seek the eigenvalue with the smallest real part in absolute value [36]. According to the definition of cumulants, $\lambda(\chi)$ can be indicated as follows:

$$\lambda(\chi) = \frac{1}{t}\sum_{k=1}^{\infty} C_k \frac{(i\chi)^k}{k!}. \tag{3.11}$$

Then the above equation is substituted into formula $|L_\chi - \lambda(\chi)I| = 0$, where $I$ denotes identity matrix. We expand its determinant and the $k$th cumulant $C_k$ is carried out by way of letting the coefficient of $(i\chi)^k$ be zero because of arbitrary $i\chi$.

# 4. Suppression of shot noise

In this work, we will reveal the numerical results due to the complex expressions of the density matrix elements derived. Specific expressions of density matrix elements with spin–orbit coupling are presented in the electronic supplementary material where the full counting statistical method is introduced with a simple model. The tunnelling rates, $k_BT$, the voltages and energies are all measured in the unit of meV throughout the paper. Figure 1 shows the value of shot noise is gradually smaller with the spin–orbit coupling increasing when the level spacing $\delta\varepsilon = \varepsilon_2 - \varepsilon_1 = 0.2\Gamma$, the spin polarization $p = 0.5$ and all transport channels participate in the electron transport simultaneously for $V = 3$. The value of shot noise is gradually decreasing when spin–orbit coupling strength increases. The suppression of shot noise can be observed from figure 1. Moreover, we can discover that super-Poisson behaviour is more obvious when spin–orbit coupling strength is smaller. This behaviour can be induced according to the

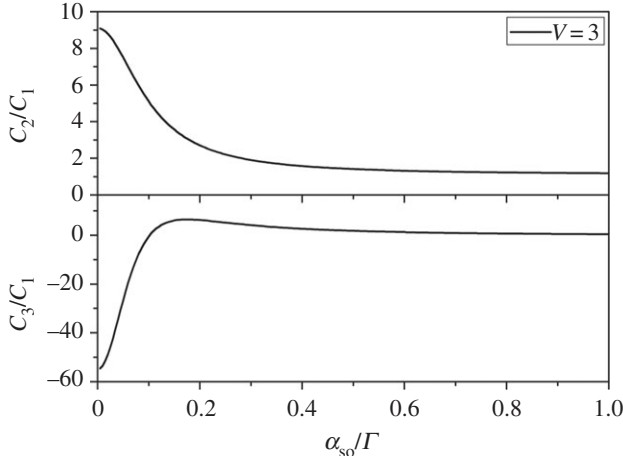

**Figure 1.** Shot noise and skewness versus spin–orbit coupling strength under the fixed bias voltage in a two-level QD system. Other parameters are $V = 3$, $\Gamma_L = \Gamma_R = \Gamma = 0.001$, the level spacing $\delta\varepsilon = \varepsilon_2 - \varepsilon_1 = 0.2\Gamma$, the spin polarization $p = 0.5$ and $k_BT = 0.02$.

following interpretations. Spin–orbit coupling interaction provides a new path for electrons tunnelling out of the QD. An electron related to spin tunnelling into the left electrode can change its spin polarization when it tunnels out of the QD arriving at the right electrode. A spin-up electron can tunnel faster than a spin-down electron due to the lead polarization, which can be explained in terms of the tunnelling rate $\Gamma_{L\uparrow(\downarrow)} = \Gamma_L(1 \pm p)$ and $\Gamma_{R\uparrow(\downarrow)} = \Gamma_R(1 \pm p)$ for the parallel lead configuration. Obviously, spin-up tunnelling rates are greater than that of spin-down according to

$$\frac{\Gamma_L^{\uparrow}}{\Gamma_L^{\downarrow}} = \frac{1+p}{1-p} \quad \text{and} \quad \frac{\Gamma_R^{\uparrow}}{\Gamma_R^{\downarrow}} = \frac{1+p}{1-p},$$

for the polarization $p > 0$, and the relationship between spin-up and spin-down tunnelling can be observed more intuitively in [40]. Besides, a spin-down electron which tunnels out with difficulty can be compelled to stay in QD for a longer time. Thus, a spin-up electron tunnelling into left lead can hardly leave dot due to the change of spin polarization caused by spin–orbit coupling interaction. However, a spin-down electron tunnelling into the left lead has a considerable likelihood of leaving the QD. The reduced tunnelling can be compensated owing to spin–orbit coupling interaction [41]. The spin block effect can be more evident when spin–orbit coupling strength is numerically small. As a result, super-Poisson noise becomes more obvious. Furthermore, we can also observe that the skewness varies to a noticeable degree and the value changes significantly from a large negative value to a small positive value.

Figure 2 indicates the great impact of polarization on shot noise and skewness when the leads are in the parallel configuration. We can see the spin polarization has a great impact on shot noise and skewness. The growth of the polarization will increase the tunnelling rates of spin-up electrons. However, it will decrease the tunnelling rates of spin-down electrons. Based on this, the spin-up current will increase and the spin-down current will decrease. Therefore, the overall current of the system which comprises the sum of the spin-up and spin-down current will not be affected. In addition, strong Coulomb interaction makes the double occupancy state of QD not be allowed, which will induce a competition between the spin-up electrons tunnelling processes and the spin-down electrons tunnelling processes. The tunnelling process for spin-up electrons is fast compared with that with spin-down electrons because of spin polarization. Most important of all, super-Poisson noise emerges clearly for relatively large polarization, which is the result of spin-dependent bunching of tunnelling events. If a spin-down electron occupies either level, the spin-up electron cannot flow through the dot in the existence of strong Coulomb blockade [42]. The spin-up electron has a possibility of tunnelling until the spin-down electron arrives at the right lead through the dot. Apart from this, we can still obtain the suppression of shot noise, and super-Poisson noise is quite evident for relatively low levels of spin–orbit coupling strength. For skewness, we can see clearly that corresponding value is gradually increasing for relatively small spin–orbit coupling strength. Nevertheless, while the skewness value is decreasing for relatively high levels of spin–orbit coupling strength.

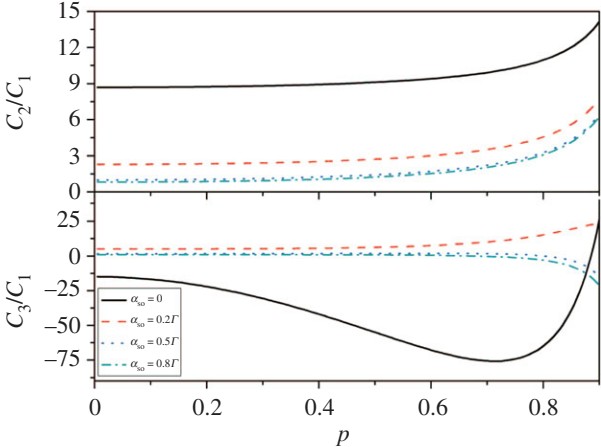

**Figure 2.** Shot noise and skewness versus the spin polarization in a two-level QD system with different $\alpha_{so}$. Other parameters are $\Gamma_L = \Gamma_R = 0.001$, the level spacing $\delta\varepsilon = \varepsilon_2 - \varepsilon_1 = 0.2\Gamma$, the bias voltage $V = 3$ and $k_BT = 0.02$.

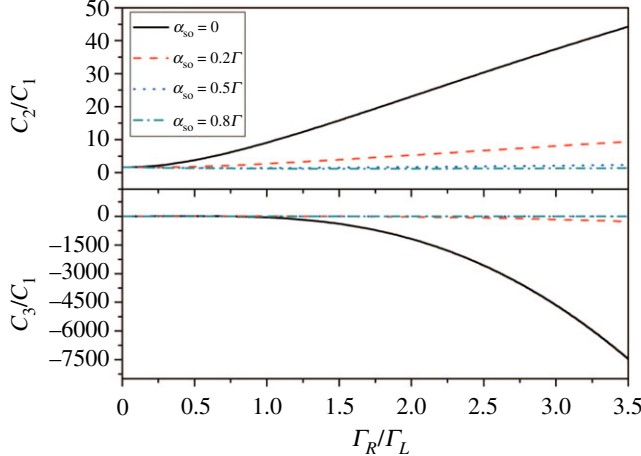

**Figure 3.** Shot noise and skewness versus the asymmetry of the levels coupling with electrode in a two-level QD system with different $\alpha_{so}$. Other parameters are $\Gamma_l = \Gamma = 0.001$, the level spacing $\delta\varepsilon = \varepsilon_2 - \varepsilon_1 = 0.2\Gamma$, the spin polarization $p = 0.5$, the bias voltage $V = 3$ and $k_BT = 0.02$.

Except for this, asymmetry of the coupling has quite an important effect on the generation of super-Poisson noise, especially when spin–orbit coupling strength is very small, as shown in figure 3. The increase of $\Gamma_R/\Gamma_L$ can lead to $\Gamma_{1L} > \Gamma_{2L}$, and $\Gamma_{1R} \gg \Gamma_{2R}$ where effective fast-and-slow channels are developed, which can result in bunching effect and induce super-Poisson noise [43]. In addition, for skewness, corresponding value changes considerably for a small positive value to a large negative value. In fact, it is found that the skewness can provide more profound information than the shot noise from the point of view of counting statistics of quantum optics.

In the above results, we focus on the spin–orbit coupling effect and do not consider the interdot tunnelling. If interdot tunnelling is considered, the quantum dot Hamiltonian is described as $\hat{H}_D = \sum_{\sigma j} \varepsilon_j d^\dagger_{j\sigma} d_{j\sigma} + \sum_\sigma i\sigma\alpha_{so}(d^\dagger_{2\sigma}d_{1\sigma} - H.c) + \sum_\sigma t(d^\dagger_{2\sigma}d_{1\sigma} + d^\dagger_{1\sigma}d_{2\sigma}) + \sum_{\sigma\sigma'} Un_{1\sigma}n_{2\sigma'}$, where $t$ denotes interdot tunnelling hopping. The eigenvalues and eigenstates of system are obtained as follows: $|\psi_{00}\rangle = |0, 0\rangle$, $E_{0,0} = 0$; $|\psi_{\sigma\sigma'}\rangle = |\sigma, \sigma'\rangle$, $E_{\sigma,\sigma'} = \varepsilon_1 + \varepsilon_2 + U$ for the non and double occupations. The one occupation states are shown as $|\psi_6\rangle = a_1|\uparrow, 0\rangle + c_1|0,\uparrow\rangle$, $|\psi_7\rangle = b_2|\downarrow, 0\rangle + c_2|0,\downarrow\rangle$, the corresponding eigenvalues $E_{6,7} = \frac{\varepsilon_2 + \varepsilon_1 - \sqrt{(\varepsilon_2 - \varepsilon_1)^2 + 4(t^2 + \alpha_{so}^2)}}{2}$; $|\psi_8\rangle = a_3|\uparrow, 0\rangle + c_3|0,\uparrow\rangle$, $|\psi_9\rangle = b_4|\downarrow, 0\rangle + c_4|0,\downarrow\rangle$, the eigenvalues $E_{8,9} = (\varepsilon_2 + \varepsilon_1 + \sqrt{(\varepsilon_2 - \varepsilon_1)^2 + 4(t^2 + \alpha_{so}^2)})/2$. Where $a_1 = b_2 = \frac{x}{\sqrt{x^2+1}}$, $a_3 = b_4 = y/\sqrt{y^2+1}$, $c_1 = c_2 = 1/\sqrt{x^2+1}$, $c_3 = c_4 = \frac{1}{\sqrt{y^2+1}}$, $x = (\varepsilon_2 - \varepsilon_1 + \sqrt{(\varepsilon_2 - \varepsilon_1)^2 + 4(t^2 + \alpha_{so}^2)})/2(i\alpha_{so} - t)$, $y = (\varepsilon_2 - \varepsilon_1 - \sqrt{(\varepsilon_2 - \varepsilon_1)^2 + 4(t^2 + \alpha_{so}^2)})/2(i\alpha_{so} - t)$.

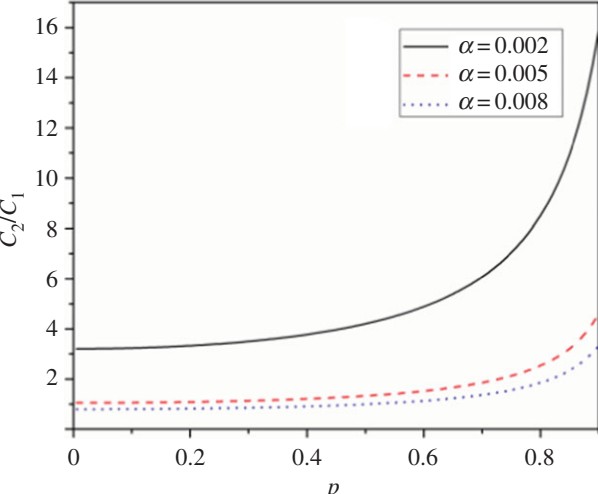

**Figure 4.** Shot noise versus the spin polarization in a two-level QD system including interdot tunnelling with different $\alpha_{so}$. Other parameters are interdot tunnelling hopping $t = 0.005$, $\Gamma_L = \Gamma_R = 0.001$, the energy level $\varepsilon_1 = \varepsilon_2 = 1$, the bias voltage $V = 2$ and $k_BT = 0.02$.

According to the numerical analysis, the same conclusion can be obtained. With spin–orbit coupling increasing, we discover the noise reduction within the range of selected parameters from figure 4.

## 5. Conclusion

On the whole, the transport properties of the spin–orbit coupled QD system are discussed under strong Coulomb blockade which can also result in super-Poisson characteristics. Besides, super-Poisson behaviour is more obvious due to the spin majority when the spin polarization is gradually increasing. Here, we assume that the spin-up electron is the spin majority. Additionally, the imbalance of the tunnelling path can give rise to super-Poisson noise. Most importantly, as is shown in figures 1–3, the spin–orbit coupling plays a significant role in determining the transport properties of a QD system. We can discover the suppression of shot noise induced by spin–orbit coupling. The super-Poisson noise can be easily observed for very small spin–orbit coupling strength, which provides us a method for exploring spin–orbit coupling existing in mesoscopic systems. We can deduce spin–orbit coupling strength from a measured noise spectrum (the second-order cumulant) by the value of the shot noise $F_a$ due to the successful measurement of the higher-order cumulants in the experiments [44–49].

Data accessibility. Our data are provided as electronic supplementary material.

Authors' contributions. Z.W. participated in the design of the study, calculated the data, participated in data analysis and drafted the manuscript; L.M. collected information; N.X. participated in data analysis; W.L. helped draft the manuscript. All authors gave final approval for publication.

Competing interests. We declare we have no competing interests.

Funding. National Natural Science Foundation of China, grant nos. 11747057, 11847111, 11747053 and Scientific and Technologial Innovation Programs of Higher Education Institutions in Shanxi Province, China, grant no. 2020L0526.

Acknowledgements. This work was supported in part by the National Natural Science Foundation of China, under grant nos. 11747057, 11847111, 11747053 and Scientific and Technologial Innovation Programs of Higher Education Institutions in Shanxi Province, China, grant no. 2020L0526.

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
