## [Peer Review File · Royal Society Open Science]

Review History

RSOS-201432.R0 (Original submission)

Review form: Reviewer 1

Is the manuscript scientifically sound in its present form?

Yes

Are the interpretations and conclusions justified by the results?

Yes

Is the language acceptable?

Yes

Do you have any ethical concerns with this paper?

No

Have you any concerns about statistical analyses in this paper?

No

Recommendation?

Accept with minor revision (please list in comments)

Comments to the Author(s)

1) The authors claim that the spin polarisation has a positive relationship with the tunnelling rate. Can the authors visualise the relationship by a plot for both spin up and spin down electrons, for example.

2) Typo: Line 58, left column, Page 3: (An) electron...

3) Type: Line 20, right column, Page 3: impact on (shot) noise...

Review form: Reviewer 2**Is the manuscript scientifically sound in its present form?**

No

Are the interpretations and conclusions justified by the results?

Yes

Is the language acceptable?

No

Do you have any ethical concerns with this paper?

No

Have you any concerns about statistical analyses in this paper?

No

Recommendation?

Reject

Comments to the Author(s)

The manuscript concerns the suppression of shot noise in a quantum dot coupled to two leads ('left' and 'right'), with spin-orbit coupling in the dot playing a key role. Based on a quantum master equation (QME) approach, a mathematical framework is presented to obtain the cumulant generating function (CGF) describing the number of electrons transferred to the right lead during a given time. From the CGF, the authors numerically compute the shot noise and the skewness for the electron transfer statistics and find that the shot noise gets suppressed with increasing spin-orbit coupling strength, which constitutes the main finding of the manuscript.

I find the topic of the manuscript interesting and timely, especially since spintronics is a vibrant field of physics that has attracted a lot of attention over the past few decades. Furthermore, I believe that the manuscript has the potential to contribute with further insights in the field.

However, I am deeply concerned about the lack of accuracy and clarity in how the model and the results are presented, and for the reasons specified below I can therefore not support publication of the manuscript:

1) I believe that, e.g., Eq. (5) is not correct. To me, it is for instance unclear what the expectation value is referring to. In any case, the authors should provide a reference for this equation.

2) Many variables and quantities are not introduced/specified in a precise way. For instance, the meanings of k and σ around Eq. (1)-(4) are not specified anywhere as far as I can see. Even if a person in the field most likely understands what the variables stand for, it should not be up to the reader to guess. In Eq. (8), it should be clarified that the CGF is F itself, not the exponential of F . In Eq. (14), λ is defined as the (!) eigenvalue of the Liouvillian, but the Liouvillian has several eigenvalues! Only one of them is the relevant eigenvalue, but the way the text is currently formulated this fact is ambiguous. These are just three examples of many. The whole manuscript needs to be recast.

3) Some of the conclusions are ambiguous too. For instance, I do not understand the sentence "We can discover the suppression of shot noise with the growth of spinorbit coupling strength as well as the decreasing value of the spinorbit coupling strength, the more apparent super-Poisson noise is, which can be observed from Fig.1."

Finally, I think it is a great idea to include supplemental material to provide the reader with additional information about the derivations of the technical results. However, the current supplemental material is presented in a completely unreadable way, with basically no explanatory text. This makes it basically impossible for me, or any other reader, to reproduce the calculations.

Based on this, I believe that the manuscript does not meet basic academic standards on accuracy, clarity and reproducibility, and I am therefore left with no other choice than to advise against publication of the manuscript.

Review form: Reviewer 3

Is the manuscript scientifically sound in its present form?

Yes

Are the interpretations and conclusions justified by the results?

Yes

Is the language acceptable?

Yes

Do you have any ethical concerns with this paper?

No

Have you any concerns about statistical analyses in this paper?

No

Recommendation?

Accept with minor revision (please list in comments)

Comments to the Author(s)

The authors study electron transport in a double quantum dot circuit in the presence of spin-orbit coupling. In particular, current, noise, and skewness are numerically calculated in the framework

of full counting statistics via a particle-resolved master equation. Their main result is that the spin-orbit coupling can lead to a reduction of the shot noise. The manuscript seems sound and interesting but the English writing could be improved. Before publication, the authors should address the following points.

1) While the effect of the spin-orbit interaction is discussed, interdot tunneling is neglected which usually is present in double quantum dot systems. It may under certain circumstances also contribute to the noise reduction as is for instance studied in Phys. Rev. B 86, 115452. Please shortly discuss, which experimental realization have you in mind.

2) Which additional insight gives the skewness beyond the Fano factor about the spin-orbit coupling? In Fig. 1, it has a maximum about $\alpha_{\text{SO}}=0.15\Gamma$. Does this maximum shift proportionally with the detuning which is for the given plot of the same order? Furthermore, the sign of the skewness seems to depend in Fig. 2 for the fully polarized case ($p=1$) on the spin-orbit coupling. Is there a simple explanation for the change of the sign?

3) In the conclusion it is stated that "The super-Poisson noise can be easily observed for greatly small spin-orbit coupling strength, which provides us a method of exploring spin-orbit coupling existing in mesoscopic systems." Is there a way to deduce its coupling strength from a measured noise spectrum? Please comment on it.

Minor comments and suggestions.

- The supplementary material should be referred to in the main text and be properly described.
- Please provide a reference for Eq. (5).
- On page 4 in the left column at lines 13-15 it is stated "[...] with a super-Poisson by $F > 1$, and a sub-Poisson by $F < 1$ ". However, F is already used for the cumulant generating function. Please fix the notation.
- In Eq. (13) occurs a "=". Please check it.
- On page 4 in the right column at lines 18 and 20 is written "short noise" instead of "shot noise". Please change it.
- In Fig. 1, a voltage and temperature are specified but from the main text it is not apparent where they enter in the formalism. Please define the correlation functions involved in Eq. (7).
- Please provide units for the tunneling rates, $k_B T$, and the voltages.

Decision letter (RSOS-201432.R0)

Dear Dr Wang

The Editors assigned to your paper RSOS-201432 "Suppression of shot noise in a spin-orbit coupled quantum dot" have now received comments from reviewers and would like you to revise the paper in accordance with the reviewer comments and any comments from the Editors. Please note this decision does not guarantee eventual acceptance.

We invite you to respond to the comments supplied below and revise your manuscript. Below the referees' and Editors' comments (where applicable) we provide additional requirements.

Final acceptance of your manuscript is dependent on these requirements being met. We provide guidance below to help you prepare your revision.

Please submit your revised manuscript and required files (see below) no later than 21 days from today's (ie 06-Jan-2021) date. Note: the ScholarOne system will 'lock' if submission of the revision is attempted 21 or more days after the deadline. If you do not think you will be able to meet this deadline please contact the editorial office immediately.

on behalf of Dr Pietro Cicuta (Associate Editor) and Miles Padgett (Subject Editor)
openscience@royalsociety.org

Associate Editor Comments to Author (Dr Pietro Cicuta):

Associate Editor: 1

Comments to the Author:

Three reviewers have commented on the manuscript, with reviewer 2 suggesting some possible serious errors, and the other reviewers more positive.

Reviewer comments to Author:

Reviewer: 1

Comments to the Author(s)

1) The authors claim that the spin polarisation has a positive relationship with the tunnelling rate. Can the authors visualise the relationship by a plot for both spin up and spin down electrons, for example.

2) Typo: Line 58, left column, Page 3: (An) electron...

3) Type: Line 20, right column, Page 3: impact on (shot) noise...

Reviewer: 2

Comments to the Author(s)

The manuscript concerns the suppression of shot noise in a quantum dot coupled to two leads ('left' and 'right'), with spin-orbit coupling in the dot playing a key role. Based on a quantum

master equation (QME) approach, a mathematical framework is presented to obtain the cumulant generating function (CGF) describing the number of electrons transferred to the right lead during a given time. From the CGF, the authors numerically compute the shot noise and the skewness for the electron transfer statistics and find that the shot noise gets suppressed with increasing spin-orbit coupling strength, which constitutes the main finding of the manuscript.

I find the topic of the manuscript interesting and timely, especially since spintronics is a vibrant field of physics that has attracted a lot of attention over the past few decades. Furthermore, I believe that the manuscript has the potential to contribute with further insights in the field.

However, I am deeply concerned about the lack of accuracy and clarity in how the model and the results are presented, and for the reasons specified below I can therefore not support publication of the manuscript:

1) I believe that, e.g., Eq. (5) is not correct. To me, it is for instance unclear what the expectation value is referring to. In any case, the authors should provide a reference for this equation.

2) Many variables and quantities are not introduced/specified in a precise way. For instance, the meanings of k and σ around Eq. (1)-(4) are not specified anywhere as far as I can see. Even if a person in the field most likely understands what the variables stand for, it should not be up to the reader to guess. In Eq. (8), it should be clarified that the CGF is F itself, not the exponential of F . In Eq. (14), λ is defined as the (!) eigenvalue of the Liouvillian, but the Liouvillian has several eigenvalues! Only one of them is the relevant eigenvalue, but the way the text is currently formulated this fact is ambiguous. These are just three examples of many. The whole manuscript needs to be recast.

3) Some of the conclusions are ambiguous too. For instance, I do not understand the sentence "We can discover the suppression of shot noise with the growth of spinorbit coupling strength as well as the decreasing value of the spinorbit coupling strength, the more apparent super-Poisson noise is, which can be observed from Fig.1."

Finally, I think it is a great idea to include supplemental material to provide the reader with additional information about the derivations of the technical results. However, the current supplemental material is presented in a completely unreadable way, with basically no explanatory text. This makes it basically impossible for me, or any other reader, to reproduce the calculations.

Based on this, I believe that the manuscript does not meet basic academic standards on accuracy, clarity and reproducibility, and I am therefore left with no other choice than to advise against publication of the manuscript.

Reviewer: 3

Comments to the Author(s)

The authors study electron transport in a double quantum dot circuit in the presence of spin-orbit coupling. In particular, current, noise, and skewness are numerically calculated in the framework of full counting statistics via a particle-resolved master equation. Their main result is that the spin-orbit coupling can lead to a reduction of the shot noise. The manuscript seems sound and interesting but the English writing could be improved. Before publication, the authors should address the following points.

1) While the effect of the spin-orbit interaction is discussed, interdot tunneling is neglected which usually is present in double quantum dot systems. It may under certain circumstances also

contribute to the noise reduction as is for instance studied in Phys. Rev. B 86, 115452. Please shortly discuss, which experimental realization have you in mind.

2) Which additional insight gives the skewness beyond the Fano factor about the spin-orbit coupling? In Fig. 1, it has a maximum about $\alpha_{\text{SO}}=0.15\Gamma$. Does this maximum shift proportionally with the detuning which is for the given plot of the same order? Furthermore, the sign of the skewness seems to depend in Fig. 2 for the fully polarized case ($p=1$) on the spin-orbit coupling. Is there a simple explanation for the change of the sign?

3) In the conclusion it is stated that "The super-Poisson noise can be easily observed for greatly small spin-orbit coupling strength, which provides us a method of exploring spin-orbit coupling existing in mesoscopic systems." Is there a way to deduce its coupling strength from a measured noise spectrum? Please comment on it.

Minor comments and suggestions.

- The supplementary material should be referred to in the main text and be properly described.
- Please provide a reference for Eq. (5).
- On page 4 in the left column at lines 13-15 it is stated "[...] with a super-Poisson by $F > 1$, and a sub-Poisson by $F < 1$ ". However, F is already used for the cumulant generating function. Please fix the notation.
- In Eq. (13) occurs a "=". Please check it.
- On page 4 in the right column at lines 18 and 20 is written "short noise" instead of "shot noise". Please change it.
- In Fig. 1, a voltage and temperature are specified but from the main text it is not apparent where they enter in the formalism. Please define the correlation functions involved in Eq. (7).
- Please provide units for the tunneling rates, kBT , and the voltages.

===PREPARING YOUR MANUSCRIPT===

If you have been asked to revise the written English in your submission as a condition of publication, you must do so, and you are expected to provide evidence that you have received language editing support. The journal would prefer that you use a professional language editing service and provide a certificate of editing, but a signed letter from a colleague who is a native

speaker of English is acceptable. Note the journal has arranged a number of discounts for authors using professional language editing services (<https://royalsociety.org/journals/authors/benefits/language-editing/>).

===PREPARING YOUR REVISION IN SCHOLARONE===

<https://royalsociety.org/journals/authors/author-guidelines/#supplementary-material> to include a suitable title and informative caption. An example of appropriate titling and captioning may be found at https://figshare.com/articles/Table_S2_from_ls_there_a_trade-

off_between_peak_performance_and_performance_breadth_across_temperatures_for_aerobic_sc
ope_in_teleost_fishes_/3843624.

Author's Response to Decision Letter for (RSOS-201432.R0)

See Appendix A.

RSOS-201432.R1 (Revision)

Review form: Reviewer 1

Is the manuscript scientifically sound in its present form?

Yes

Are the interpretations and conclusions justified by the results?

Yes

Is the language acceptable?

Yes

Do you have any ethical concerns with this paper?

No

Have you any concerns about statistical analyses in this paper?

No

Recommendation?

Accept as is

Comments to the Author(s)

I accept the author's response and the revisions as it is.

Review form: Reviewer 2

Is the manuscript scientifically sound in its present form?

No

Are the interpretations and conclusions justified by the results?

Yes

Is the language acceptable?

No

Do you have any ethical concerns with this paper?

No

Have you any concerns about statistical analyses in this paper?

No

Recommendation?

Accept with minor revision (please list in comments)

Comments to the Author(s)

I do not think my questions have been properly addressed. The supplemental material looks indeed much better, but there are still some ambiguities in the main text. First, I think it is still unclear what the expectation value in Eq. (3.1) is computed with respect to. The expectation value needs to be defined in the text. Second, in connection to Eq. (3.10), it should be clarified, in a well-defined way, which eigenvalue λ_1 corresponds to (I presume it is the one with the largest real part). Overall I think the manuscript has improved substantially compared to the first review round, but the language throughout the manuscript may also be polished a bit further.

Review form: Reviewer 3

Is the manuscript scientifically sound in its present form?

Yes

Are the interpretations and conclusions justified by the results?

Yes

Is the language acceptable?

Yes

Do you have any ethical concerns with this paper?

No

Have you any concerns about statistical analyses in this paper?

No

Recommendation?

Accept with minor revision (please list in comments)

Comments to the Author(s)

Before assessing the revised manuscript, I would like to indicate that its present form is missing the figures which has to be fixed. Anyway, I am assuming that figures 1-3 are the same as in the previous version and that Fig.4 is the one occurring in the reply to Reviewer 1. Moreover, the supplementary material has some formatting issues in the paragraph above Eq. (1) and below Eq. (3).

Apart from these technical points, I would suggest the manuscript for publication in the Royal Society Open Science since the authors have satisfactorily addressed all my questions and provided with the new Fig.4 and its description additional value to their work.

Decision letter (RSOS-201432.R1)

Dear Dr Wang

On behalf of the Editors, we are pleased to inform you that your Manuscript RSOS-201432.R1 "Suppression of shot noise in a spin-orbit coupled quantum dot" has been accepted for publication in Royal Society Open Science subject to minor revision in accordance with the referees' reports. Please find the referees' comments along with any feedback from the Editors below my signature.

Please submit your revised manuscript and required files (see below) no later than 7 days from today's (ie 23-Feb-2021) date. Note: the ScholarOne system will 'lock' if submission of the revision is attempted 7 or more days after the deadline. If you do not think you will be able to meet this deadline please contact the editorial office immediately.

on behalf of Dr Pietro Cicuta (Associate Editor) and Miles Padgett (Subject Editor)
openscience@royalsociety.org

Associate Editor Comments to Author (Dr Pietro Cicuta):

Associate Editor: 1

Comments to the Author:

There are still improvements to be made, following reviewers 2 and 3, but it is converging to a manuscript that can be accepted.

Reviewer comments to Author:

Reviewer: 1

Comments to the Author(s)

I accept the author's response and the revisions as it is.

Reviewer: 3

Comments to the Author(s)

Before assessing the revised manuscript, I would like to indicate that its present form is missing the figures which has to be fixed. Anyway, I am assuming that figures 1-3 are the same as in the previous version and that Fig.4 is the one occurring in the reply to Reviewer 1. Moreover, the supplementary material has some formatting issues in the paragraph above Eq. (1) and below Eq. (3).

Apart from these technical points, I would suggest the manuscript for publication in the Royal Society Open Science since the authors have satisfactorily addressed all my questions and provided with the new Fig.4 and its description additional value to their work.

Reviewer: 2

Comments to the Author(s)

I do not think my questions have been properly addressed. The supplemental material looks indeed much better, but there are still some ambiguities in the main text. First, I think it is still unclear what the expectation value in Eq. (3.1) is computed with respect to. The expectation value needs to be defined in the text. Second, in connection to Eq. (3.10), it should be clarified, in a well-defined way, which eigenvalue λ_1 corresponds to (I presume it is the one with the largest real part). Overall I think the manuscript has improved substantially compared to the first review round, but the language throughout the manuscript may also be polished a bit further.

===PREPARING YOUR MANUSCRIPT===

===PREPARING YOUR REVISION IN SCHOLARONE===

Author's Response to Decision Letter for (RSOS-201432.R1)

See Appendix B.

Decision letter (RSOS-201432.R2)

Dear Dr Wang

On behalf of the Editors, we are pleased to inform you that your Manuscript RSOS-201432.R2 "Suppression of shot noise in a spin-orbit coupled quantum dot" has been accepted for publication in Royal Society Open Science subject to minor revision in accordance with the referees' reports. Please find the referees' comments along with any feedback from the Editors below my signature.

Please submit your revised manuscript and required files (see below) no later than 7 days from today's (ie 15-Mar-2021) date. Note: the ScholarOne system will 'lock' if submission of the revision is attempted 7 or more days after the deadline. If you do not think you will be able to meet this deadline please contact the editorial office immediately.

on behalf of Dr Pietro Cicuta (Associate Editor) and Miles Padgett (Subject Editor)

Associate Editor Comments to Author (Dr Pietro Cicuta):

Associate Editor

Comments to the Author:

It looks like the authors have addressed referee comments. On my reading of this however, I see that the abstract needs improving - it should follow a more standard format of stating what sort of work this is (experimental, theoretical, computational), what is the open question, what has been done and what has been discovered.

===PREPARING YOUR MANUSCRIPT===

===PREPARING YOUR REVISION IN SCHOLARONE===

Author's Response to Decision Letter for (RSOS-201432.R2)

See Appendix C.

Decision letter (RSOS-201432.R3)

Dear Dr Wang,

It is a pleasure to accept your manuscript entitled "Suppression of shot noise in a spin-orbit coupled quantum dot" in its current form for publication in Royal Society Open Science.

on behalf of Dr Pietro Cicuta (Associate Editor) and Miles Padgett (Subject Editor)
openscience@royalsociety.org

Appendix A

Response to the comments of Referees and Editors

Reviewer: 1

Comments to the Author(s)

1) The authors claim that the spin polarisation has a positive relationship with the tunnelling rate. Can the authors visualise the relationship by a plot for both spin up and spin down electrons, for example.

Response: This can be explained in terms of the tunneling rate $\Gamma_{L\uparrow(\downarrow)} = \Gamma_L(1 \pm p)$

and $\Gamma_{R\uparrow(\downarrow)} = \Gamma_R(1 \pm p)$ for the parallel lead configuration. Obviously, spin-up

tunneling rates are greater than that of spin-down according to $\frac{\Gamma_L^\uparrow}{\Gamma_L^\downarrow} = \frac{1+p}{1-p}$,

$\frac{\Gamma_R^\uparrow}{\Gamma_R^\downarrow} = \frac{1+p}{1-p}$ for the polarization $p > 0$ and the relationship plot between spin-up and

spin-down electrons tunneling can be observed more intuitively from Fig.4 of reference {*Physical Review B* 75, 165303(2007)} as follows:

2) Typo: Line 58, left column, Page 3: (An) electron...

Response: "A" is corrected to "An" in the revised version.

3) Typo: Line 20, right column, Page 3: impact on (shot) noise...

Response: "short noise" is corrected to "shot noise" in the revised version.

Reviewer: 2

Comments to the Author(s)

1) I believe that, e.g., Eq. (5) is not correct. To me, it is for instance unclear what the expectation value is referring to. In any case, the authors should provide a reference for this equation.

Response: We add the reference{Phys. Rev. B, 2005, 71, 205304} about Eq. (5) { Eq. (3.1) of the revised version}.

2) Many variables and quantities are not introduced/specified in a precise way. For instance, the meanings of k and σ around Eq. (1)-(4) are not specified anywhere as far as I can see. Even if a person in the field most likely understands what the variables stand for, it should not be up to the reader to guess. In Eq. (8), it should be clarified that the CGF is F itself, not the exponential of F . In Eq. (14), λ is defined as the (!) eigenvalue of the Liouvillian, but the Liouvillian has several eigenvalues! Only one of them is the relevant eigenvalue, but the way the text is currently formulated this fact is ambiguous. These are just three examples of many. The whole manuscript needs to be recast.

Response: We add the meanings of k and σ around Eq. (1)-(4) { Eq. (2.1-2.4) of the revised version}. The description about the cumulant generating function(CGF) is corrected. The description about $\lambda_1(\chi)$ is corrected. $\lambda_1(\chi)$ is one of eigenvalues of the Liouvillian operator L_χ . According to the definition of

cumulants, $\lambda_1(\chi)$ can be written as $\lambda_1(\chi) = \frac{1}{t} \sum_{k=1}^{\infty} C_k \frac{(i\chi)^k}{k!}$, thus inserting this

into the secular equation $|L_\chi - \lambda_1(\chi)I| = 0$, here I denotes identity matrix.

Moreover, we add the definition of other variables and quantities which are not mentioned in previous manuscript.

3) Some of the conclusions are ambiguous too. For instance, I do not understand the sentence “We can discover the suppression of shot noise with the growth of spin orbit coupling strength as well as the decreasing value of the spinorbit coupling strength, the more apparent super-Poisson noise is, which can be observed from Fig.1.”

Response: The conclusion is corrected to "The value of shot noise is gradually decreasing when spin orbit coupling strength increases. The suppression of shot

noise can be observed from Fig.1. Moreover, we can discover that super-Poisson behaviour is more obvious when spin orbit coupling strength is smaller."

Finally, I think it is a great idea to include supplemental material to provide the reader with additional information about the derivations of the technical results. However, the current supplemental material is presented in a completely unreadable way, with basically no explanatory text. This makes it basically impossible for me, or any other reader, to reproduce the calculations.

Response: The supplemental material is recast. We add the calculation process of full counting statistics with a simple model and solve the eigenvalues and eigenstates of Hamiltonian of the main text and the density matrix element with explanatory text.

Reviewer: 3

Comments to the Author(s)

1) While the effect of the spin-orbit interaction is discussed, interdot tunneling is neglected which usually is present in double quantum dot systems. It may under certain circumstances also contribute to the noise reduction as is for instance studied in Phys. Rev. B 86, 115452. Please shortly discuss, which experimental realization have you in mind.

Response: We add the description and Fig.4 about interdot tunneling before Conclusion in the revised version. We can obtain the noise reduction with spin orbit coupling strength increasing within the range of selected parameters.

2) Which additional insight gives the skewness beyond the Fano factor about the spin-orbit coupling? In Fig. 1, it has a maximum about $\alpha_{\text{SO}}=0.15\Gamma$. Does this maximum shift proportionally with the detuning which is for the given plot of the same order? Furthermore, the sign of the skewness seems to depend in Fig. 2 for the fully polarized case ($p=1$) on the spin-orbit coupling. Is there a simple explanation for the change of the sign?

Response: The maximum value occurs as a result of spin accumulation ($m = n_{\uparrow} - n_{\downarrow}$). $n_{\sigma} = \langle \hat{n}_{\sigma} \rangle, \hat{n}_{\sigma} = d_{\sigma}^{\dagger} d_{\sigma}$. The spin accumulation increases when p increases. Spin accumulation reaches maximum for the fully polarized case ($p=1$).

3) In the conclusion it is stated that "The super-Poisson noise can be easily observed for greatly small spin-orbit coupling strength, which provides us a method of

exploring spin-orbit coupling existing in mesoscopic systems." Is there a way to deduce its coupling strength from a measured noise spectrum? Please comment on it.

Response: Recently, there have been many experiments to measure the higher order cumulants seen in reference {*Phys. Rev. Lett.*91, 196601 (2003); *Phys. Rev. Lett.*95, 176601 (2005); *Nat. Commun.*3, 612 (2012); *Phys. Rev. Lett.* 112, 036801 (2014); *Nat. Phys.*3, 243 (2007); *Science* 312 1634 (2006)}. We can deduce its coupling strength from a measured noise spectrum (the second order cumulant) by the value of the shot noise F_a .

Minor comments and suggestions.

- The supplementary material should be referred to in the main text and be properly described.

Response: The supplementary material is referred to in the revised manuscript.

- Please provide a reference for Eq. (5).

Response: We add the reference{*Phys. Rev. B*, 2005, 71, 205304} about Eq. (5) { Eq. (3.1) of the revised version}.

- On page 4 in the left column at lines 13-15 it is stated "[...] with a super-Poisson by $F > 1$, and a sub-Poisson by $F < 1$ ". However, F is already used for the cumulant generating function. Please fix the notation.

Response: " $F > 1$, $F < 1$ " is corrected to " $F_a > 1$, $F_a < 1$ " in the revised version.

- In Eq. (13) occurs a "=". Please check it.

Response: Eq. (13) { Eq. (3.9) of the revised version} is corrected.

- On page 4 in the right column at lines 18 and 20 is written "short noise" instead of "shot noise". Please change it.

Response: "short noise" is corrected to "shot noise" in the revised version.

- In Fig. 1, a voltage and temperature are specified but from the main text it is not apparent where they enter in the formalism. Please define the correlation functions involved in Eq. (7).

Response: The correlation functions are defined under Eq. (7) { Eq. (3.3) of the revised version}.

- Please provide units for the tunneling rates, $k_B T$, and the voltages.

Response: We add the description about units in section {Suppression of shot

noise} in the revised version. The tunneling rates, $k_{\{B\}T}$, the voltages and energies are all measured in the unit of meV throughout the paper.

Appendix B

Response to the comments of Referees and Editors

Reviewer: 2

Comments to the Author(s)

I do not think my questions have been properly addressed. The supplemental material looks indeed much better, but there are still some ambiguities in the main text. First, I think it is still unclear what the expectation value in Eq. (3.1) is computed with respect to. The expectation value needs to be defined in the text. Second, in connection to Eq. (3.10), it should be clarified, in a well-defined way, which eigenvalue λ_1 corresponds to (I presume it is the one with the largest real part). Overall I think the manuscript has improved substantially compared to the first review round, but the language throughout the manuscript may also be polished a bit further.

Response: We add the definition of the expectation value in Eq. (3.1)

$$\left\langle L(t)\tilde{G}(t,t')L(t')\tilde{G}^\dagger(t,t') \right\rangle = \text{Tr}_B \left[L(t)\tilde{G}(t,t')L(t')\tilde{G}^\dagger(t,t')\rho_B \right] \quad \text{with } \rho_B \text{ the}$$

density matrix of the electron reservoirs in the revised version. The eigenvalues of the matrix L_χ defined in Eq. (3.10) have the expansion

$$\lambda = \lambda_0 + \lambda_1\chi + \lambda_2\chi^2 + O(\chi^3). \quad \text{We seek the eigenvalue with the smallest real part in}$$

absolute value. That eigenvalue has $\lambda_0 = 0$. We also express the eigenvector w corresponding to λ and the matrix itself in a power series in χ ,

$$w = w_0 + w_1\chi + w_2\chi^2 + O(\chi^3), \quad L = L_0 + L_1\chi + L_2\chi^2 + O(\chi^3). \quad \text{Inserting the above}$$

expansions into the eigenvalue equation $Lw = \lambda w$ yields the following relationships of, respectively, zero, first, and second order:

$$L_0w_0 = 0, \quad L_1w_0 + L_0w_1 = \lambda_1w_0, \quad L_2w_0 + L_1w_1 + L_0w_2 = \lambda_2w_0 + \lambda_1w_1. \quad \text{The coefficients } L_k$$

are known, while w_k and λ_k remain to be found by solving these equations sequentially. The first two cumulants then follow from $C_1 = -it\lambda_1$, $C_2 = -2t\lambda_2$.

In an analog way it is possible to calculate higher cumulants *{Physical Review B 74, 125315 (2006)}*. The language throughout the manuscript is polished a bit

further in bold text in the revised version.

Reviewer: 3

Comments to the Author(s)

Before assessing the revised manuscript, I would like to indicate that its present form is missing the figures which has to be fixed. Anyway, I am assuming that figures 1-3 are the same as in the previous version and that Fig.4 is the one occurring in the reply to Reviewer 1. Moreover, the supplementary material has some formatting issues in the paragraph above Eq. (1) and below Eq. (3).

Apart from these technical points, I would suggest the manuscript for publication in the Royal Society Open Science since the authors have satisfactorily addressed all my questions and provided with the new Fig.4 and its description additional value to their work.

Response: These technical points are corrected in the revised version and some formatting issues in the supplementary material are corrected.

Appendix C

Response to the comments of Referees and Editors

Associate Editor Comments to the Author:

It looks like the authors have addressed referee comments. On my reading of this however, I see that the abstract needs improving - it should follow a more standard format of stating what sort of work this is (experimental, theoretical, computational), what is the open question, what has been done and what has been discovered.

Response: The abstract is corrected in the revised version as follows:

We study theoretically the transport properties of electrons in a quantum dot system with spin-orbit coupling. By using the quantum master equation approach, the shot noise and skewness of the transport electrons are calculated. We obtain super-Poisson noise behaviour by investigating the full counting statistics of the transport system. We discover super-Poisson behaviour is more obvious with the spin polarization increasing. More importantly, we discover the suppression of shot noise induced by spin-orbit coupling. The value of shot noise is gradually decreasing when spin orbit coupling strength increases.